# Effect of 6 Months of Physical Training on the Physical Fitness of Young Brazilian Army Cadets

**DOI:** 10.3390/healthcare9111439

**Published:** 2021-10-25

**Authors:** Rafael Melo De Oliveira, Eduardo Borba Neves, Samir Ezequiel Da Rosa, Runer Augusto Marson, Rodrigo Gomes de Souza Vale, Jairo José Monteiro Morgado, Wilson de Assis Lacerda Junior, Renato Souza Pinto Soeiro, Rodolfo de Alkmim Moreira Nunes

**Affiliations:** 1Brazilian Army Physical Fitness Research Institute, Brazilian Army (IPCFEx/EB), Rio de Janeiro 22291-090, Brazil; samirrosa@eb.mil.br (S.E.D.R.); prof2.divpesq@ipcfex.eb.mil.br (R.A.M.); ch.fisiologia@ipcfex.eb.mil.br (J.J.M.M.); adj2.fisiologia@ipcfex.eb.mil.br (W.d.A.L.J.); dir@ipcfex.eb.mil.br (R.S.P.S.); 2Graduate Program of Biomedical Engineering, Federal Technological University of Paraná (PPGEB/UTFPR), Curitiba 80230-901, Brazil; eduardoneves@utfpr.edu.br; 3Graduate Program in Exercise and Sport Sciences, Rio de Janeiro State University (PPCEE/UERJ), Rio de Janeiro 20550-013, Brazil; rodrigo.vale@estacio.br (R.G.d.S.V.); rodolfo.alkmin@labees.net (R.d.A.M.N.)

**Keywords:** military, physical training, muscle fitness, cardiorespiratory fitness, DXA, bone

## Abstract

Following the increase in the employment of women in conflicts around the world, the federal government of Brazil enacted a law which determines the participation of women in the military. The aim of this study was to analyze the effect of six months of physical training (PT) on the physical fitness of young Brazilian Army cadets to carry out the physical assessments provided in military training. Sixty-eight members of the (19.4 ± 1.0 years) military from the Brazilian Army (BA), with BMI of (23.61 ± 2.17/21.81 ± 2.26) respectively and divided in two groups (men/women) participated in the study. PT was conducted by Manual EB20-MC10.350. Anthropometric measurements and assessment of body composition by dual X-ray absorptiometry were performed. The Student’s *t* test, percentage evolution equation, and Levene test were used. Results showedasignificant increase in anthropometric variables and cardiorespiratory fitness in both groups. Bone health variables and visceral fat presented a significant increase in the malegroup. In terms of muscle fitness handgrip and isometric strength there was no significant variation between the groups and push-up and pull-up there was significant variation between the groups. Percentage evolution was greater in female group. The conclusion shows PT was able to cause beneficial changes, promoting positive improvement in bone health, especially in women. Also, PT was shown to enhance cardiorespiratory capacity, and muscle fitness of the upper limbs in all participants.

## 1. Introduction

Throughout the history of military conflicts between the world military powers, policies have been established to exclude female involvementin real military operations with in areas with high physical demand, such as in the American Armed Forces, due to physiological differences in the performance of the main components of fitness physical activity and the high incidence of musculoskeletal injuries [1].

Even withthese exclusionary policies, women have been more employed in recent war conflicts because of the changing nature of the battlefield, in which the majority of military specialties, regardless of sex are engaged against the enemy [2].

Following the increase in the useof women in conflicts around the world through the military forces of the main powers, the federal government of Brazil enacted Federal Law No. 12705 on 8 August 2012, which determines the insertion of women in the Military Occupation specialties (MOS) with directly engage the enemy of the Armed Forces of Brazil within a maximum period of 5 years [3].

Regarding the insertion of women in the Brazilian Army (LEMB), the Army Physical Training Research Institute (IPCFEx) initially carried out a non-experimental scientific literature review with the main military powers in the world to identify and establish the main physical tests to be used in the Initial Physical Fitness Exam (EAFI) and in physical assessments throughout military training, in agreement with the main physiological differences in the components of physical fitness and income proportion between men and women [1,4,5].

There are studies that show physiological differences between males and females in the main components of physical fitness, such as body composition, in which females have 12% more body fat, 50% less lean body mass and 30% less of muscle mass in the lower limbs. Regarding cardiorespiratory capacity, females have less than 15 to 30compared to men, which directly reflects on the performance of physical activities, negatively influencing the training for military tasks where the demand for physical capacity in certain situations is high [1,4,5].

In this scenario, it is important to monitor the physical profile, achieved through military physical training, throughout the military training of young people and women, especially women, with a view to adequate performance in the physical assessments of the main components of physical fitness which provide support for the training of military tasks with high physical demands. In this sense, the objective of this study was to analyze the evolution of the anthropometric profile, bone mass, and physical fitness of young adults of both sexes, under the influence of regular physical training, to achieve adequate physical fitness during this period in the training course for Brazilian Army officers.

## 2. Materials and Methods

### 2.1. Study Design and Population

This research was a randomized, uncontrolled, longitudinal and comparative clinical trial. It was carried out at the Army Preparatory School of Cadets, in 2017, with the first class containing female students. This study was approved by the Ethical Committee in Research according to approval code: 55948016.1.0000.5289 and the approval date of 30 March 2016. All participants signed an informed consent form.

The present study obeys the demand of the total sample size (*n* = 54) calculated by the G*Power 3.1.9.7 software with effect size (0.5), α err prob (0.05) and Power (1-β Err Prob) (0.95).

The sample consisted of twogroups:a group composed of physically active young women (*n* = 31) with average age (19.5 ± 1.1) and average height (164.28 ± 3.79), and another group with physically active young men (*n* = 37) of average age (19.5 ± 1.3) and average height (175.6 ± 5.0).

In this study, an initial assessment was performed before regular physical training of the physical fitness variables described in the manuscript and an assessment at the end of the training period.

The variances between groups defined ashomogeneous by the Levene’s test (*p* = 0.193), and there was no significant difference in mean age between the groups by the independent *t* test (*p* = 0.961). The selection of the male and female group was random and all members of the sample were volunteers.

For inclusion criteria, the sample participants were volunteers, qualified by the doctor to carry out the training and perform all the tests provided for in the study. Those who were not involved in at least 70% of them were excluded from the study due to injury or clinical health problems of the training sessions or did not participate in all the moments of tests foreseen in the study. Drug intake was not controlled throughout the training period and the menstrual cycle was not controlled in the training period and in the evaluations.

### 2.2. Methodological Procedures

#### Physical Training

The improvement of performance in the main components of physical fitness followed the recommendations of the manual EB 20-MC10.350 Military Physical Training for the prescription and progression of training. The training period lasted for sixmonths, with two performance peaks at the end of the two 12-week macrocycles.

The initial test was performed at week zero and the retest at the end of the training period. On average there were fourtraining sessions per week lasting 90 min, the goal being mainly to improve cardiorespiratory fitness, muscle fitness, body composition and flexibility.

The two macrocycles were divided into basic and specific phases with peak performance at the end of each macrocycle according to the linear evolution acquired, with the projection of the highest peak performance at the end of the second macrocycle.

A.Cardiorespiratory training

Regarding cardiorespiratory running training (Table 1), an average of three to four training sessions per week were carried out, with the duration of 30 to 50 minfor each session according to the established intensity.

The intensity of the running cardiorespiratory training sessions were classified by the percentage training zones of the heart rate reserve [4]. In the last fourweeks of each macrocycle, there was an increase in the intensity of the training sessions in order to improve performance in this component of physical fitness.

B.Muscle Fitness Training

Muscle fitness training was carried out with a frequency of threesessions per week in general, with two sessions with an emphasis on shoulder and arm exercises aimed at the evolution of performance in push-up and pull-up tests and one session of resistance circuit training with shoulder, knee, leg, arm and hip exercises, performing two sets in each circuit exercise with load resisted at 50% of 1RM [4,6].

C.Flexibility training

Active static stretching was performed at least threetimes a week, with the main muscle groups involved in the training session up to the point of maximum tension with slight discomfort, maintaining this angle for at least 30 s. The exercises used were lateral bending of the trunk to extend the dorsal musculature, raising the arms to the rear of the trunk to provoke the extension of the pectoral muscles, flexing the legs with the feet close to the buttocks to extend the anterior thigh muscles (quadriceps), adduction of the legs sitting on the floor to extend the adductor muscles of the legs, flexion of the trunk sitting on the floor to provoke stretching of the posterior muscles of the thigh, leaning the trunk forward for stretching of the muscles of the lower back, extension of the leg backwards to extend the ilio psoas muscle, lifting the front of the feet with one leg forward with the knee bent and the other leg with the knee extended to extend the gastrocnemius muscles.

### 2.3. Evaluation Protocols and Instruments

#### Anthropometry

In relation to anthropometric indicators, the sample members were assessed for height, weight, body mass index, waist circumference, hip circumference and waist/hip ratio. The weight was obtained using the G TECH GLASS electronic scale with a 100 g precision scale and a maximum weight of 150 kg.

Height was assessed by the STANDARD SANNY stadiometer with a precision of 1 mm and a measurement range of 80 to 220 cm.

The body mass index was found using the formula (Weight (kg)/(height (m)^2^). Waist and hip circumference were obtained using an anthropometric body measuring tape TL 200 TEKLIFE, graduated in centimeters with a maximum height of 200 cm. The waist/hip ratio was performed using the formula (waist circumference (cm)/circumference of the hip (cm)) [5,6].

An anthropometric assessment was conducted by two physical education professionals from the IPCFEx following the protocol standardized by international standards for anthropometric assessment (ISAK) [7].

Anthropometric variables were measured by the same experienced evaluator. The technical errors of measurement intra-evaluator were considered acceptable, being 0.97% [8].

### 2.4. Body Composition

The body composition assessment was performed using DXA, iLunar model, from GE Healthcare (GE Healhcare, Madison, WI, USA), with the enCore 2015 software (version 14.10.022). Before each acquisition a DXA scanner was calibrated according to the manufacturer’s instructions [9,10,11]. Furthermore, the calibration column phantom was applied weekly. From the scan of the whole body, the data of total fat mass (FM), total lean mass (LM), percentage of total body fat (%BF-DXA) and the fat mass index (FMI), FM (kg)/Height^2^ (m^2^), were obtained. VAT was measured and a region-of-interest was automatically defined whose caudal limit was placed at the top of the iliac crest and its height was set to 20% of the distance from the top of the iliac crest to the base of the skull to define itscephalad limit [9,10,11]. VAT mass (g) was automatically transposed into volume (cm^3^) using a constant correction factor (0.94 g/mL). The measurements in the calibration block (daily) presented acceptable coefficients of variation 1.0% [9,10,11].

### 2.5. Physical Fitness

The 3000-m field test was performed, as a cardiorespiratory test, on a 400-m track, where each individual ran this distance in the shortest possible time with the possibility of calculating the peak speed in the protocol, caloric expenditure and metabolic rate of effort. Maximum repetition tests of the arms flexion on the fixed bar and flexion of the arms on the floor were performed, having to perform the maximum number of repetitions until the momentary muscle failure to assess the muscular fitness of the upper limbs.

Muscle fitness of the lower limbs was assessed using a back-leg-chest dynamometers, in which the subjects performed three maximum attempts at the peak of the isometric strength of the legs with an interval of onemin between attempts.

The handgrip strength was assessed using a handgrip dynamometer, with which the members of the sample made three attempts to grasp the dynamometer with maximum contraction, and an interval of onemin between attempts. Flexibility was assessed by the sit and reach test on the Wells bench [12,13]. The menstrual cycle was not controlled in the training period and in the evaluations.

### 2.6. Statistical Analysis

For the statistical analysis, raw variable values, percentage values and percentage evolution values were used. The percentage evolution (Evol%) was calculated using the equation [((Ass2/Ass1) × 100) − 100]. (Ass1 = Assessment 1/Ass 2 = Assessment 2).

In this study, the normality test (Kolmogorov Smirnov) was conducted to guide the statistical analysis. As a result of the groups being *n* > 30, the parametric T test with samples in pairs was performed on all variables of the main components of physical fitness measured in the referred study. (*p* ≤ 0.05) was used.

Regarding the comparison between the training efficiency groups in anthropometric variables, body composition, bone mass and physical performance variables, the Levene test for homogeneity of variances and the Independent Samples *t* Test were performed.

## 3. Results

Table 2 shows the results of anthropometric variables, body composition and bone mass at the beginning and at the end of the training cycle. According to the results of the anthropometric variables, it can be seen that in both groups there was a significant increase in the mean values of waist circumference (WC), hip circumference (HC), and body mass index (BMI). Regarding the waist-to-hip ratio (WHR), there was a significant increase only for the group with young female adults.

Regarding the variables of body composition, it can be noted that there was a significant increase in the average values of total mass for both groups under the influence of training. There was no significant difference in both groups in terms of fat mass, and in relation to lean mass there was an increased significant difference in the mean values only for the female group.In the variable percentage of total fat (% F total), there was a significant increase in the mean values only in the group with young adult males.

Concerning the effect size on variables related to body composition, it is noted that CQ and BMD were classified as “very small” (d = 0.01–0.19), WHR as “small” (d = 0.2–0.49), Fat Mass “Large” (d = 0.8–1.19), WC, Lean Mass were rated very large (d = 1.2–1.9) and BMI, % F Total was classified “Huge” (d ≥ 2.0) [11].

Concerning the results of the bone mass variables, in the total bone mass content (BMC) there was a significant increase in the mean values only for the male group. In relation to BMC in the regions of the arm, trunk and legs in both groups, there was a significant increase in mean values. Regarding the bone mass density (BMD) variable, only the group with young female adults showed a significant increase in mean values.

In the variables visceral adipose tissue (VAT) and visceral fat, a significant decrease was observed in the average values in the group of young adult males; on the other hand, the result in the female group was reversed with a significant increase in the average values of these two variables. The Z-Score variable in both groups under the effect of training showed a similar behavior with a significant increase in mean values.

In the variables related to bone mass and visceral adipose tissue, it can be seen that VAT was classified in relation to the effect size as “very small” (d = 0.01–0.19) and Trunk BMC as “Very Large” (d = 1.2–1.9).The variables Total BMC, Leg BMC and Arms BMC received the classification “Huge” (d ≥ 2.0) [11].

Regarding the variables of physical performance in the main components of physical fitness, Table 3 describes the results of the evolution of cardiorespiratory fitness, muscle fitness of the upper and lower limbs and the flexibility of the hamstring muscles.

Concerning the variables of cardiorespiratory fitness under the effect of the training period, it is noted that both groups significantly increased the average performance values. In the variable referring to flexibility, it was noticed that in both groups there was no significant increase or decrease in the average values from the initial evaluation to the final one.

With regard to muscle fitness, in the variables of handgrip and isometric strength of the legs, there was no significant variation in the mean values in both groups. In the variables of muscle fitness for push-up and pull-up, a significant increase in the average values of performance between evaluations under the effect of physical training was noticed in both groups.

As foreffect size of the performance variables described in Table 3, BW received the classification “large” (0.8–1.19), Misll was classified as “very large” (1.2–1.99) and the variables Average Speed, RHG, LHG and Pull-up had the classification “Huge” (d ≥ 2.0) [11].

Table 4 shows the comparison, between male and female, of the effectiveness of training in anthropometric variables, body composition, bone mass and physical performance in the main components of physical fitness. It is observed that there was a significant difference in training efficiency between male and female in the variables VAT, Average Speed, pull-up and push-up, with the male sex showing better results.

In these variables that had a significant difference in the effectiveness of physical training, referring to the effect size, it can be seen that the variables VAT and Average Speed were classified as “Large” (0.80–1.19). The Push-Up variable received the classification “Very Large” and the Pull-Up variable was classified as “Huge” (d ≥ 2.0) [11].

## 4. Discussion

The present study sought to determine the evolution of the anthropometric profile and body composition through bone mineral density, bone mass and physical performance of the main components of physical fitness of young male and female military adults under the influence of 24 weeks of physical training and routine military activities.

Regarding anthropometric variables, an increase in mean waist circumference (WC) values was observed for both sexes (*p* < 0.001), hip circumference (QC) for men (*p* < 0.001) and women (*p* = 0.008), and body mass index (BMI) for males (*p* < 0.001) and females (*p* < 0.001). There was a significant increase in the mean values of total mass (TM) for both sexes (*p* < 0.001), while in the variable% G total there was only an increase with a significant difference for males (*p* = 0.017) compared to females (*p* = 0.630), and in the fat mass variable, there was no significant difference in male (*p* = 0.232) and female (*p* = 0.286). The increases in the mean values of anthropometric variables (WC/CQ/BMI/TM) in both sexes and in the percentage of fat (% F) for males only are contrary to what was observed in studies with young adults exposed to regular physical training [14,15].

In the VAT and visceral F variables, there was a significant decrease in the average values of males (*p* = 0.016) and a significant increase in females (*p* = 0.003). These results corroborate with studies that identified the greatest reduction in visceral fat in males compared to females exposed to regular physical training [16].

Another aspect that underlies the results found for this variable in this study is that the greater reduction in visceral fat has an important relationship with the higher initial values of fat in this male body region [17]. In relation to the significant increase in VAT and visceral F with females, studies show that the decrease in this variable is related to adecrease in body mass, waist circumference and body mass index. The inverse behavior of females in this study, with the increase in mean values of body mass, may explain the significant increase in VAT.

When analyzing the percentage evolution of the male VAT variable (Evol (%) = −17.96 ± 61.66) compared to that forfemales (Evol (%) = 59.46 ± 110.56), it is noted that there was asignificant decrease in male visceral fat (*p* < 0.001) in relation to female, showing that the training had better benefits for the male sex, which is in alignment in with studies that evaluated the behavior of this variable between the sexes [16,17].

Regarding lean mass, there is a significant increase in females (*p* < 0.001) compared to males (*p* = 0.737), with the influence of physical training and functional activities with high physical demand. The evolution of this variable for females agrees with studies that demonstrate an increase in lean mass under the influence of regular cardiorespiratory fitness training and especially muscle fitness training [18,19].

On the other hand, the workload of the training period does not seem to have been sufficient to significantly evolve this variable with the male sex. It was observed that the functional activities and the same physical training for both sexes cooperated more effectively for the evolution of the female lean mass, because the female effort to perform the task was probably greater due to the physiological differences between the men [20]. In agreement with the greater effort of women in relation to men when they carry out activities with the same external load, according to Nindl, Jones, Van Arsdale, Kelly and Kraemer [1], young adult males haveapproximately 50% more muscle mass of the upper limbs and 30% more mass lower limb muscle.

It is noticed that in the BMD variable the behavior was the same in relation to lean mass, in males and females, with the increase of the significant mean value only in females (*p* < 0.001), and the males remained in the same parameters (*p* = 0.458). These findings reinforce the evolution of muscle fitness in line with the increase in lean mass, bone mass and bone mineral density reported in studies [18,21].

Regarding the BMC variable, in the body segments evaluated in this study, it is noted that there was a significant evolution with the increase in the mean values in the trunk and arms for both sexes (*p* < 0.001) for the leg segment, the same behavior was found for male (*p* = 0.005) and female (*p* = 0.007).

These results are supported by studies that report that muscle fitness training promotes increased strength, muscle mass, BMD, BMC and bone resistance to physical stress. Such benefits are important to prevent bone loss throughout life [18,20,21,22].

In the sit-and-reach test using the BW, the mean values of male (24.25 ± 7.17) and female (30.39 ± 7.30) at the final moment of evaluation of the study received the classification “needs improve” and “fair”; according to classification parameters established by a Canadian study [4], there was no significant increase under the influence of the period of physical training and functional activities of the average BW values in the male group (*p* = 0.285) or in the female group (*p* = 0.643).

The results of this study can be explained by analyzing the training load over the entire period. The prescription of flexibility training throughout the training period was not enough for this muscle group, as the study [18] recommend performing a weekly frequency of ≥two to three times, 10 to 30 s, stretching the main muscles two to four times for each muscle, to bring about beneficial changes.

Regarding RHG and LHG, the behavior was similar to BW without significant increase in the average values of male (*p* = 0.826/0.295) and female (*p* = 0.056/0.150). These results are similar to the results of studies [12] that report small changes in different training modes that do not specifically address handgrip strength.

Concerning cardiorespiratory fitness using the 3000-m running protocol, it is noted that the initial and final performance of the male and femalesthrough the variable average speed in the running protocol, were (244.19 ± 10.61 m/min), (188.52 ± 11.08 m/min) and (254.01 ± 7.77 m/min), (208.81 ± 10.22 m/min), respectively. Regarding the variable average oxygen consumption in the initial and final evaluation of the male and female sex, the results were VO2 Effort M1 men = 52.33 mL/kg/min (Percentile = 70), VO2 Effort M1 women = 41.20 mL/kg/min (Percentile = 55) and VO2 Effort M2 men = 54.30 mL/kg/min (Percentile = 75), VO2 Effort M2 women = 45.26 mL/kg/min (Percentile = 75) [4]. It is noted that there was a significant evolution in a beneficial way in the mean values of cardiorespiratory capacity yield for both sexes (*p* < 0.001).

The behavior of studies carried out with young military adults [23] and non-military [24] corroborate the results found in this study in which the range of performance in the initial evaluation through the average oxygen consumption of the male and female was (40.70–55.6 mL/kg/min) and (31.10–41.20 mL/kg/min) respectively, and in the final evaluation it was (46.70–60.2 mL/kg/min) and (33.40–46.80 mL/kg/min), respectively.

Regarding the training efficiency of this variable in this study, it is clear that the female gender (9.80 ± 9.74%) presented the average percentage values above significantly in relation to the male gender (3.70 ± 5.02%) evidencing the greatest impact of training on female cardiorespiratory evolution.

In the same studies highlighted above, the amplitude of the percentage increase in performance under the influence of the training period for males and females was (4.40–17.20%) and (7.30–18.30%), respectively. The results of the current study in terms of this variable partially agree with the trend found in the aforementioned studies, as the female gender presented the result in the percentage range of income and the male gender was below the lower limit.

Regarding the variable maximum isometric strength of lower limbs (MISLL), by means of an analog dynamometer that measures low back and leg strength, it was observed that the average initial and final values of male performance (257.58 ± 81.19/235.36 ± 95.88) and female performance (152.65 ± 47.78/148.67 ± 38.77) respectively are found in higher parameters of the male income range (132.5–207.3 kgf) and female (63.4–126.62 kgf) identified in other studies [25] that evaluated the same variable. In agreement with the superior performance of this study in this variable, the study by [26] classifies the performance of this variable for both sexes in the “EXCELLENT” parameter.

Regarding the evolution of the performance of this variable under the influence of regular physical training, it can be noted that there was no significant difference in the average values of males (*p* = 0.215) and females (*p* = 0.533). These results can be explained through studies [12] that identify the need for specific training in the valence for the adequate evolution in performance.

Concerning the push-up variable, one of the physical assessments that measures the muscular fitness of the upper limbs, it can be seen in this study that there was a significant increase in the average values of performance between the initial and final assessment of the male gender (32.55 ± 7.16/38.52 ± 10.36 maximum repetitions) and the female gender (14.68 ± 4.46/21.02 ± 4.67 maximum repetitions), respectively. It is inferred that the training period and the training prescription over the period was adequate to promote this evolution [18].

Regarding the percentage evolution under the influence of the regular physical training period of the FBS variable, there was a greater significant evolution in the female gender (Evol% = 46.12 ± 42.94) in relation to the male (Evol% = 23.50 ± 19.52).

To the detriment of the higher percentage evolution of the female sex, the average result of the initial (14.68 ± 4.46) and final (21.02 ± 4.67) performance of the female in relation to the initial (32.66 ± 7.24) and final (39.69 ± 7.93) males in this physical evaluation, point us to a higher male income compared to female with the average income ratio between the sexes (53%) as studies direct in the scientific literature [1].

Regarding the percentage efficiency under the influence of the training period identified in the current study with male (23.50%) and female (46.12%) when comparing with the range of percentage efficiency of other studies with male (41–62%) and female (23–160%), it can be seen that the percentage evolution of the male gender in the current study was below the range of the other studies and the female gender had an evolution within the range of the other studies. The percentage evolution of the current study, especially of the male sex, can be explained because the initial levels of average income are in high parameters of classification for the age group [27] and close to the maximum requirements necessary for the formation, and for this reason they can have influenced the smallest evolution over the training period [28].

Referring to the pull-up variable that also measures the muscular fitness of the upper limbs, the average performance obtained in this study in the initial evaluation of male (8.78 ± 3.13) and female (0.48 ± 1.16) and in the final evaluation male and female, respectively (10.09 ± 2.98), (3.25 ± 3.36), demonstrated that the training period caused significant beneficial changes in the average results in males (*p* < 0.001) and females (*p* < 0.001).

Regarding the percentage evolution of income, it should be noted that although the evolution was greater for females (Evol% = 1771.47 ± 2380.01) compared to males (Evol (%) = 19.43 ± 39.07), the performance throughout the study was significantly lower for females compared to males from the beginning to the end, showing great difficulty for females to evolve performance in this physical evaluation [29]. These findings reinforce the need for further studies to better understand the female performance in this variable and to optimize physical training strategies to approximate the maximum performance of this variable between the sexes.

A limitation of the study was the impossibility of controlling the participants’ diet, but through the collection of information with members of the sample about food intake, it strengthened the assumption that eating disorders contributed to these results. This behavior is similar to a trend in recent decades among young military adults of both sexes with problems of overweight and obesity, related or not to eating disorders, under the influence of psychological stress common to military training [30].

## 5. Conclusions

We can conclude that the period of regular physical training in relation to body composition in both sexes was not able to cause beneficial changes with the decrease in body fat. Regarding bone mass and bone mineral density, exercise promoted positive improvement, especially with females.

Regarding physical performance in the main components of physical fitness, it was noticed that the training was able to evolve the cardiorespiratory capacity and muscle fitness of the upper limbs, but did not promote changes in the muscular fitness of the lower limbs and the flexibility of the hamstrings.

It is inferred in future studies to increase awareness and monitoring of good practices of healthy diet to better assist the beneficial effects of body composition under the influence of regular physical exercise.

Regarding the improvement of physical fitness, more research needs to be carried out in terms of training approaches aimed at better improving the muscular fitness of female upper limbs and the muscular fitness of the lower limbs and flexibility in both sexes.

## Figures and Tables

**Table 1 healthcare-09-01439-t001:** Regarding cardiorespiratory running training.

Week	Session	Intensity	Volume
1st to 4th week			
03 running exercise	02 running exercise	Moderate and vigorous	volume of 30 to 45 min
	01 Interval training exercise	Vigorous	04 a 08 laps of 400 m at supra-maximum speed
9th to 12th week	02 running exercise	Moderate and vigorous	Volume of 25 to 45 min
	02 Interval training exercise	Vigorous	08 to 12 laps of 400 m at supra-maximum speed
13th to 20th week	02 running exercise	Moderate and vigorous	volume of 30 to 45 min
	01 Interval training exercise	Vigorous	08 to 12 laps of 400 m at supra-maximum speed
21th to 24th week	02 running exercise	Moderate and vigorous	Volume of 25 to 45 min
	02 Interval training exercise	Vigorous	08 to 12 laps of 400 m at supra-maximum speed

**Table 2 healthcare-09-01439-t002:** Effect of training on anthropometric variables, body composition and bone mass.

Parameters	MALE (*n* = 37)			FEMALE (*n* = 31)					
	Ass 1	Ass 2	*p*	Ass 1	Ass 2	*p*	Effect Size Ass 2	CI 95%	
								Lower	Higher
WC (cm)	77.13 ± 4.59	78.82 ± 4.81	<0.001 *	68.54 ± 4.54	69.75 ± 4.88	0.008 *	1.87	1.30	2.44
CQ (cm)	93.43 ± 4.83	95.69 ± 4.49	<0.001 *	92.92 ± 5.36	95.66 ± 4.56	<0.001 *	0.01	−0.47	0.48
BMI (kg/m^2^)	22.97 ± 2.14	23.61 ± 2.17	<0.001 *	21.30 ± 2.20	21.81 ± 2.26	0.001 *	2.17	1.57	2.77
WHR	0.82 ± 0.31	0.82 ± 0.33	0.384	0.73 ± 0.02	0.72 ± 0.03	0.384 *	0.41	−0.07	0.89
Total Mass (kg)	71.09 ± 7.88	73.09 ± 7.86	<0.001 *	57.41 ± 6.48	59.03 ± 6.76	<0.001 *	1.91	1.33	2.48
Fat Mass (kg)	11.89 ± 3.83	12.40 ± 4.11	0.232	15.53 ± 3.12	15.87 ± 3.23	0.286	−0.93	−1.43	−0.43
Lean mass (kg)	56.35 ± 5.71	55.78 ± 10.77	0.737	39.56 ± 4.03	40.80 ± 4.01	<0.001 *	1.78	1.22	2.35
% F total	16.50 ± 4.37	17.25 ± 3.75	0.017 *	26.88 ± 3.32	26.70 ± 3.01	0.630	−2.75	−3.42	−2.09
Total BMC (g)	3022.95 ± 318.83	3067.16 ± 320.61	<0.001 *	2313.42 ± 281.84	2335.75 ± 291.71	0.205	2.38	1.75	3.00
Leg BMC (g)	1188.16 ± 157.49	1194.95 ± 154.85	0.005 *	838.33 ± 106.18	850.72 ± 101.86	0.007 *	2.58	1.94	3.23
Trunk BMC (g)	865.00 ± 112.12	885.95 ± 112.21	<0.001 *	672.61 ± 107.58	686.53 ± 103.98	<0.001 *	1.84	1.27	2.41
Arms BMC (g)	434.95 ± 50.53	444.54 ± 54.03	<0.001 *	285.36 ± 36.17	295.75 ± 33.20	<0.001 *	3.25	2.53	3.98
BMD (mg/cm^2^)	1.23 ± 0.85	1.25 ± 0.83	0.458	1.12 ± 0.95	1.13 ± 0.86	0.001 *	0.14	−0.34	0.62
VAT (g)	0.29 ± 0.22	0.18 ± 0.12	0.016 *	0.07 ± 0.07	0.11 ± 0.08	0.003 *	0.13	−0.35	0.61
Z Score	0.60 ± 0.80	0.65 ± 0.11	0.439	0.65 ± 1.05	0.81 ± 0.96	0.013 *	−0.25	−0.72	0.23

WC = waist circumference; CQ = hip circumference; BMI = body mass index; WHR = waist-to-hip ratio; Total% F = percentage of total fat; BMC = bone mass content; BMD = bone mass density; VAT = visceral adipose tissue; Visceral F = visceral fat; * *p* < 0.05 (Student *t*).

**Table 3 healthcare-09-01439-t003:** Effect of training on physical fitness components.

Parameters	MALE (*n* = 37)			FEMALE (*n* = 31)					
	Ass 1	Ass 2	*p*	Ass 1	Ass 2	*p*	Effect Size Ass 2	CI 95%	
								Lower	Higher
Average Speed (m/min) (3000 m)	244.19 ± 10.61	254.01 ± 7.77	<0.001 *	188.52 ± 11.08	208.81 ± 10.22	0.000 *	5.04	4.07	6.01
BW (cm)	23.37 ± 7.66	24.25 ± 7.17	0.285	30.66 ± 7.31	30.39 ± 7.30	0.643	−0.85	−1.35	−0.35
RHG(kgf)	46.68 ± 7.21	46.88 ± 8.56	0.826	30.86 ± 4.47	32.07 ± 5.48	0.056	2.02	1.44	2.61
LHG (kgf)	45.36 ± 8.08	46.14 ± 8.33	0.295	30.50 ± 4.57	31.37 ± 6.35	0.150	1.97	1.39	2.55
Pull Up (Rep Máx)	8.78 ± 3.13	10.09 ± 2.98	<0.001 *	0.48 ± 1.16	3.25 ± 3.36	0.000 *	2.17	1.57	2.77
Push-Up (Rep Máx)	32.55 ± 7.16	38.52 ± 10.36	<0.001 *	14.68 ± 4.46	21.02 ± 4.67	0.000 *	2.12	1.52	2.71
MISLL (kgf)	257.58 ± 81.19	235.36 ± 95.88	0.215	152.65 ± 47.78	148.67 ± 38.77	0.533	1.15	0.64	1.67

Average speed = average speed; BW = bank of wells; RHG = right hand grip; LHG = left hand grip; MISLL = maximum isometric strength of lower limbs. * *p* < 0.05 (Student *t*).

**Table 4 healthcare-09-01439-t004:** Comparison, between males and females, of training effectiveness in anthropometric variables, body composition, bone mass and physical performance in the main components of physical fitness.

Parameters	Male (%)	Female (%)	Teste de Levene (*p* Value)	Teste T Independente (*p* Value)	Effect Size	CI 95%	
						Lower	Higher
Evol % WC	2.22 ± 2.76	1.80 ± 3.84	0.23	0.59	0.13	−0.35	0.61
Evol % BMI	2.83 ± 3.69	2.46 ± 4.15	0.41	0.69	0.09	−0.38	0.57
Evol % WHR	−0.23 ± 1.65	−1.16 ± 3.49	0.02	0.15	0.35	−0.13	0.83
Evol % Total Mass	2.90 ± 3.22	2.89 ± 4.05	0.23	0.98	0.00	−0.47	0.48
Evol % Fat Mass	9.39 ± 13.37	2.94 ± 12.29	0.46	0.03	0.50	0.02	0.98
Evol% Lean Mass	2.05 ± 2.42	3.22 ± 3.18	0.04	0.08	−0.42	−0.90	0.06
Evol% Total G	6.14 ± 10.70	−0.15 ± 9.00	0.31	0.00	0.63	0.14	1.12
Evol% Total BMC	1.48 ± 1.25	1.04 ± 4.88	0.07	0.60	0.13	−0.35	0.61
Evol% Legs BMC	0.61 ± 1.18	1.62 ± 3.90	0.21	0.13	−0.36	−0.85	0.12
Evol% Trunk BMC	2.51 ± 3.19	2.24 ± 3.14	0.92	0.71	0.09	−0.39	0.56
Evol% Arms BMC	2.18 ± 2.73	3.20 ± 4.41	0.75	0.23	−0.28	−0.76	0.20
Evol% Total BMD	1.61 ± 9.43	1.31 ± 2.31	0.00 #	0.85	0.04	−0.44	0.52
Evol %VATkg	−17.96 ± 61.66	59.46 ± 110.56	0.03 #	0.00 *	−0.89	−1.39	−0.39
Evol%ZscoreTotal	3.20 ± 62.71	0.61 ± 95.09	0.32	0.89	0.03	−0.44	0.51
Evol% Avg Speed	4.19 ± 5.02	11.31 ± 9.74	0.00	0.00	−0.94	−1.45	−0.44
Evol% BW	9.00 ± 37.32	−0.14 ± 12.99	0.11	0.18	0.32	−0.16	0.80
Evol % RHG	0.60 ± 10.98	4.10 ± 12.08	0.69	0.20	−0.30	−0.78	0.18
Evol % LHG	2.34 ± 11.06	2.49 ± 11.67	0.33	0.95	−0.01	−0.49	0.46
Evol% Pull up	19.43 ± 39.07	1771.47 ± 2380.01	0.00 #	0.00 *	−1.09	−1.60	−0.58
Evol% Push up	23.50 ± 19.52	46.12 ± 42.94	0.00 #	0.00 *	−0.70	−1.19	−0.21
Evol% MISLL	−1.92 ± 36.48	2.84 ± 32.88	0.93	0.56	−0.14	−0.61	0.34

Evol% = percentage evolution; WC = waist circumference; BMI = body mass index; WHR = waist-to-hip ratio; Total F = total fat; BMC = Bone Mass Content; BMD = Bone Mass Density; VAT = visceral adipose tissue; Visceral F = visceral fat; BW = bank of wells; RHG = right hand grip; LHG = left hand grip; MISLL = maximum isometric strength of lower limbs. * *p* < 0.05 (Student t); # *p* < 0.05 (Levene’s test).

## Data Availability

All data generated in this study are contained within this publication.

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
