# Peer review of "Effect of 6 Months of Physical Training on the Physical Fitness of Young Brazilian Army Cadets"

_healthcare, 2021, doi:10.3390/healthcare9111439_

Round 1

Reviewer 1 Report

With this study authors aimed to analyze the effect on both genders of regular physical training on anthropometric profile, bone mass, physical fitness.

The English language must be revised. Some words and phrases are written in Portuguese.

Introduction

the introduction should enlighten about what is already known about the effect of Physical training in women and men anthropometric and fitness profiles.

Methods

Please refer to the rationale for different recruitment methodology and if or how the menstrual cycle was dealt with in the women assessment.

Results

Please format table 2 that is incomplete, and explain better the values presented (eg: effect size and CI 95%) on both tables 2 and 3.

Discussion

Please refer to your explanation for the differences found in your study from others.

Since the manuscript has no line number I add my specific comments on the manuscript itself.

Author Response

REPLY LETTER TO REVIEWER 1

QUESTION 1. Regarding the comment/suggestion The English language must be revised. Some words and phrases are written in Portuguese”.

ANSWER 1: I would like to inform you that a thorough review of the English language was carried out and all errors identified in the text of the article were corrected.

QUESTION 2. Concerning about the comment/ suggestion “ the introduction should enlighten about what is already known about the effect of Physical training in women and men anthropometric and fitness profiles”.

ANSWER 2: Based on previous studies, important physiological differences between the sexes in the main components of physical fitness such as body composition, muscle fitness and cardiorespiratory capacity were discriminated in the 5th paragraph of the introduction.

QUESTION 3. About the comment/suggestion ”Please refer to the rationale for different recruitment methodology…”

ANSWER 3: I inform you that this author made a mistake in describing the recruitment of the sample composed of young female adults, which was corrected and discriminated in paragraph 5 of item 2.1 Study design and population.

QUESTION 4. Concerning about “if or how the menstrual cycle was dealt with in the women assessment”.

ANSWER 4: Agreeing with the suggestion made by the reviewer, I inform that it was inserted in paragraph 6 of item 2.1 Study design and population that “ the menstrual cycle was not controlled in the training period and in the evaluations”.

QUESTION 5. Regarding the comment/suggestion in the results “ Please format table 2 that is incomplete, and explain better the values presented (eg: effect size and CI 95%) on both tables 2 and 3”.

ANSWER 5: The format of Tables 2 and 3 has been updated and the effect size results have been better explained.

Thank you for all the suggestions made to improve this author's article.

Great Regards

Rafael Melo de Oliveira

Reviewer 2 Report

In the title, ‘evolution’ might not be a good word to describe your research question and contents, please consider using another word.

Can your study sample size to sufficient to achieve your research aims?

According to the method, it is different with research aim. Please do some necessary changes where possible and appropriate.

Table 2 was incomplete.

Based on your research question and method mainly, I think this study was inconsistent and not logical.

Author Response

REPLY LETTER TO REVIEWER 2

QUESTION 1. Regarding the comment/suggestion In the title, ‘evolution’ might not be a good word to describe your research question and contents, please consider using another word”.

ANSWER 1: In accordance with the suggestion made by the reviewer, I inform that the title was changed to “Effect of 6 months of physical training on the physical fitness of young Brazilian Army cadets”. 

QUESTION 2. Concerning about the comment/ suggestion “Can your study sample size to sufficient to achieve your research aims”?

ANSWER 2: In response to the reviewer's questioning, I inform that the sample calculation was performed and inserted in the 2nd paragraph of item 2.1 Study design and population using the G*Power 3.1.9.7 software.

QUESTION 3. About the comment/suggestion” According to the method, it is different with research aim. Please do some necessary changes where possible and appropriate”.

 ANSWER 3: The description of the research aim has been updated to better meet the understanding of the methodological design used in the research of this article.

QUESTION 4. Concerning about “Table 2 was incomplete”.

ANSWER 4: Table 2 was revised and updated according to the reviewer's comment.

QUESTION 5. Regarding the comment/suggestion “ Based on your research question and method mainly, I think this study was inconsistent and not logical”.

ANSWER 5: The description of the research question was updated in the introduction to improve consistency and rationale regarding the methodological design not identified by the reviewer.

Thank you for all the suggestions made to improve this author's article.

Great Regards

Rafael Melo de Oliveira

Reviewer 3 Report

It is an interesting topic and I congratulate the authors on the research, but indicate some comments to improve the manuscript.

The title should change the word anthropometry to body composition. To study the anthropometric profile, it is necessary to indicate data on skin folds, girth, breadth bone, etc.

Review all bibliographic references of the manuscript. It is not correct [31] [32] [33] [34] [35], it must be [31-35].

If possible, the first part of the abstract should include some aspect of anthropometry and justify the interest of the research.

In material and methods section:

  • indicate type of study
  • Measurement moment of each variable studied
  • Has any anthropometric measurement protocol been followed? the authors must indicate it.
  • Has the technical measurement error (TEM) been taken into account?
  • Have the anthropometric measurements been performed by a trained anthropometrist?
  • The authors must indicate the bibliographic references on recommendations or protocol to be followed in the measurement of the different variables studied (DXA, anthropometry).

In table 2:

  • Table 2 does not display properly.
  • % G total is Total% F, please modify
  • Add weight and height data

Page 9 lacks bibliographic references in most of the sentences. It should be compared with other similar studies.

The discussion is very long, the authors should try to shorten it. In addition, i recommend that the authors try to discuss body composition data with data obtained through anthropometry.

Author Response

REPLY LETTER TO REVIEWER 3

QUESTION 1. Regarding the comment/suggestion The title should change the word anthropometry to body composition. To study the anthropometric profile, it is necessary to indicate data on skin folds, girth, breadth bone, etc”.

ANSWER 1: In agreement with the reviewer, the title was modified with the approach based on body composition through the evaluation with the DXA, iLunar model.

QUESTION 2. Concerning about the comment/ suggestion “Review all bibliographic references of the manuscript. It is not correct [31] [32] [33] [34] [35], it must be [31-35].

ANSWER 2: In response to the reviewer's questioning, all references were reviewed and those that were in the wrong format were corrected. 

QUESTION 3. About the comment/suggestionIf possible, the first part of the abstract should include some aspect of anthropometry and justify the interest of the research”.

 ANSWER 3: In accordance with the reviewer's comment/suggestion, the variable BMI was inserted and updating the research objective we believe will clarify the justification for the research.

QUESTION 4. Concerning about the topics type of study and Measurement moment of each variable studied

ANSWER 4: the type of study and the moments in which the variables were evaluated in the study were inserted in the 1st paragraph and in the 4th paragraph of item 2.1. Study design and population of the article.

QUESTION 5. Regarding the comment/suggestion about anthropometric measurement protocol been followed and technical measurement error”.

ANSWER 5: Both topics comments were included in the item describing the protocols in the anthropometric assessment of the variables in this article.

QUESTION 6. About the comment/suggestion on the anthropometric measurements been performed by a trained anthropometrist and bibliographic references on recommendations or protocol to be followed in the measurement of the different variables studied (DXA, anthropometry)”.

ANSWER 6: Agreeing with the important comments on these topics, the anthropometry and body composition item included the necessary information.

QUESTION 7. Regarding the comment/suggestion about display of the table 2, total %F and add weight and height”.

ANSWER 7: The format of Table 2 was updated, total %F was entered, weight was measured by the total mass provided by the DXA, and height was entered in paragraph 3 of item 2.1 Study design and population.

QUESTION 8. Regarding the comment/suggestion about lacks bibliographic references in most of the sentences. It should be compared with other similar studies ”.

ANSWER 8: In the discussion, a review was made so that all results are compared with similar studies as the intention was from the beginning.

QUESTION 9. Regarding the comment/suggestion about The discussion is very long, the authors should try to shorten it. In addition, i recommend that the authors try to discuss body composition data with data obtained through anthropometry”.

ANSWER 9: In compliance with the reviewer's recommendations, 9 paragraphs were removed to reduce the discussion and the approach in this research was on body composition through the DXA assessment, as it is the gold standard.

Thank you for all the suggestions made to improve this author's article.

Great Regards

Rafael Melo de Oliveira

Round 2

Reviewer 2 Report

The authors have addressed my comments.